# Preparation and Photocatalytic Properties of a Bagasse Cellulose-Supported Nano-TiO_2_ Photocatalytic-Coupled Microbial Carrier

**DOI:** 10.3390/ma13071645

**Published:** 2020-04-02

**Authors:** Jianhua Xiong, Yinna Liang, Hao Cheng, Shuocheng Guo, Chunlin Jiao, Hongxiang Zhu, Shuangfei Wang, Jiaxiang Liang, Qifeng Yang, Guoning Chen

**Affiliations:** 1School of Resources, Environment and Materials, Guangxi University, Nanning 530004, China; happybear99@126.com (J.X.); liangyinna0754@126.com (Y.L.); m13171793629@163.com (C.J.); 2Guangxi Key Laboratory of Clean Pulp & Papermaking and Pollution Control, Nanning 530004, China; iamchenghao@126.com (H.C.); zhx@gxu.edu.cn (H.Z.); dwaasg@sina.cn (J.L.); 3Guangxi Key Laboratory of Green Processing of Sugar Resources, College of Biological and Chemical Engineering, Guangxi University of Science and Technology, Liuzhou 545006, China; 4Analysis and Testing Center, Yangtze Delta Region institute of Tsinghua University, Jiaxing 314006, China; badmrguo@hotmail.com; 5Guangxi Bossco Environmental Protection Technology Co., Ltd., Nanning 530007, China; yangqf@bossco.cc (Q.Y.); chenguonin2@126.com (G.C.)

**Keywords:** intimate coupling of photocatalysis and biodegradation, bagasse cellulose–nano TiO_2_ composite carrier, characterization, photocatalytic performance

## Abstract

Intimate coupling of photocatalysis and biodegradation (ICPB) has shown promise in removing unwanted organic compounds from water. In this study, bagasse cellulose titanium dioxide composite carrier (SBC-TiO_2_) was prepared by low-temperature foaming methods. The optimum preparation conditions, material characterization and photocatalytic performance of the composite carrier were then explored. By conducting a single factor test, we found that bagasse cellulose with a mass fraction of 4%, a polyvinyl alcohol solution (PVA) with a mass fraction of 5% and 20 g of a pore-forming agent were optimum conditions for the composite carrier. Under these conditions, good wet density, porosity, water absorption and retention could be realized. Scanning electron microscopy (SEM) results showed that the composite carrier exhibited good biologic adhesion. X-ray spectroscopy (EDS) results confirmed the successful incorporation of nano-TiO_2_ dioxide into the composite carrier. When the mass concentration of methylene blue (MB) was 10 mg L^−1^ at 200 mL, 2 g of the composite carrier was added and the initial pH value of the reaction was maintained at 6, the catalytic effect was best under these conditions and the degradation rate reached 78.91% after 6 h. The method of preparing the composite carrier can aid in the degradation of hard-to-degrade organic compounds via ICPB. These results provide a solid platform for technical research and development in the field of wastewater treatment.

## 1. Introduction

Textiles, with printing and dyeing industries, have become a pillar industry, bringing huge economic developments to China. For example, research shows that the output value of China’s textile industry has increased from 3477.94 billion yuan in 2012 to 3778.5 billion yuan in 2015, with an annual growth rate of 6%. After this, the total output value of China’s textile industry has been showing a gentle upward trend [1,2]. Meanwhile, the total amount of wastewater discharged from textile industry also continued to increase, with an average annual growth of 5.54% [3]. The characteristics of high turbidity, pH value, chromaticity and organic pollutant content while poor biodegradability [4,5,6,7] in the textile emission wastewater makes it hard for treatment, which is harmful to human beings and the environment [8].

Photocatalysis is a method to purify pollutants by using photocatalyst with redox ability under ultraviolet irradiation. And TiO_2_ is the most valuable catalyst because of its high catalytic activity, stability, low cost and corrosion resistance [9]. In 2008, a research group led by professor Marsolek [10], academician of the American Academy of Engineering, proposed for the first time a promising method for the treatment of refractory organics, namely the intimate coupling of photocatalysis and biodegradation (ICPB). In this technology, the microbe and photocatalyst are supported on a porous carrier to realize the simultaneous degradation of pollutants by photocatalytic coupling under aerobic conditions in the same reactor. ICPB system has the advantages of high recovery rate of photocatalyst and high degradation efficiency of organic matter compared with the single photocatalyst and biodegradation. Zhang (2015) made polyurethane foam plastics (PUF)/TiO_2_ composite carrier by low-temperature ultrasonic impregnation method and used for degradation of phenolic pollutants. The results showed that the treatment efficiency of ICPB was 90.33%, which was better than that of solo photocatalysis (11.87%) or microbial degradation (26.57%).

Following its proposal, ICPB technology has been successfully applied to the degradation of refractory organic compounds such as phenol [11,12,13], 2,4,5-trichlorophenol [10,14] 2, 4-diaminotoluene [15] and tetracycline [16,17] as well as printing and dyeing wastewater [18,19]. Most photocatalytic carriers are inorganic materials, such as activated carbon, zeolite, glass, ceramics, etc., which have good treatment effect [20,21]. Li G et al. [15] used TiO_2_ coated polyurethane sponge carrier to remove 2,4,5-trichlorophos (TCP), and successfully proved the feasibility and advantages of ICPB direct coupling system. Wen et al. [16] prepared porous ceramic carrier under UV light to degrade 2,4-DNT, the degradation rate reached 78%. Yongming et al. [14] designed an immobilized photocatalytic biofilm reactor based on a honeycomb ceramic carrier, which was very stable under UV light. However, these carriers have limitations, the carriers used in the above studies are not fully loaded with TiO_2_, and it is scorched by light with the extension of use time. At the same time, the preparation process of the carrier is complex, and the adsorption effect is poor, which is not conducive to the survival of the biofilm in the carrier. The complexity of the method of preparing the functional ceramic carrier made it difficult to use in practical engineering applications.

To develop an environment-friendly and biodegradable photocatalytic response composite carrier, this paper used bagasse cellulose as the basic framework. It has the advantages of simple preparation process, strong adsorption [22,23,24], high porosity [25,26] and high specific surface of materials to be used to load titanium dioxide, which is an ideal carrier for biofilm loading. We used Nano-TiO_2_ as the photocatalyst, and taking methylene blue (MB) as the target pollutant—studied the physical and chemical properties and photocatalysis of the composite carrier. The degradation efficiency of MB was also investigated to provide reference for optimizing ICPB system. This study is carried out according to the flow chart (Figure 1).

## 2. Materials and Methods

### 2.1. Chemicals

The following chemicals were employed in this study: Glutaraldehyde (analytical grade; Tianjin Damao Chemical Reagent Factory, Tianjin, China), anhydrous sodium sulfate (analytical grade; Guangdong Chemical Reagent Engineering Technology Research and Development Center, Guangdong, China), polyvinyl alcohol 1799 (analytical grade; Chengdu Kelong Chemical Reagent Factory), zinc chloride (analytical grade; Guangdong Chemical Reagent Engineering Technology Research and Development Center), nano-titanium dioxide (Liuzhou Ruosi Nanomaterials Technology Co., Ltd., Liuzhou, China), bleached bagasse (Guigang City Gui Sugar Co., Ltd. Guigang, China), MB (analytical grade; Chengdu Kelong Chemical Reagent Factory, Chengdu, China), hydrochloric acid (analytical grade; Chengdu Kelong Chemical Co., Ltd. Chengdu, China) and sodium hydroxide (analytical grade; Guangdong Chemical Reagent Engineering Technology Research and Development Center).

### 2.2. Preparation of Bagasse Cellulose Composite Carrier

We dissolved 69.12 g of ZnCl_2_ in 26.88 g deionized water, then added cellulose and stirred (80 °C for 1 h). A polyvinyl alcohol (PVA) solution was added to this to create a uniform solution that was mixed for 1 h, after which 10 mL of 5 g titanium dioxide suspension was added; this was then stirred for 30 min. Subsequently, 10 mL of 4% glutaraldehyde was added at 50 °C, and a pore-forming agent, Na_2_SO_4_, was added after continuous reaction for 30 min. After mixing for 1 h, the mixture was poured into a mold and solidified in deionized water at 50 °C for 2 days. Finally, a solidified carrier was formed after washing, refrigeration and freeze-drying. The carrier was cut into a block of about 10 mm × 10 mm × 10 mm to prepare for subsequent experiments.

### 2.3. Biofilm Cultivation

The activated sludge was obtained from Guangxi Bossco Environmental Protection Technology Co., Ltd. (Nanning, China) The experimental method was as follows: 100 mL of activated sludge was placed in a 1000-mL reactor and 5 g of the bagasse cellulose titanium dioxide composite carrier (SBC-TiO_2_) was added. Film hanging was completed after one day of stifling. The carrier was then transferred to the reactor, a certain concentration of MB solution containing the culture medium was added and aeration was continued for domestication. The water was changed once per day. After about seven days, completion of microbial domestication on the carrier. Then, the carrier was taken out for characterization.

### 2.4. Characterization

We used a scanning electron microscope (Gemini SEM 300, Zeiss, Cambridge, Britain) to visualize the structure of the composite carrier, the adhesion mode of TiO_2_ and the accumulation of biofilm. X-ray spectroscopy (EDS) was used to measure the relative concentrations of Ti and O elements in the composite carrier.

The composite carrier was immersed in deionized water at 20 °C for 24 h to determine its wet density ρ. First, we took out the soaked carrier and absorbed the surface moisture with filter paper. Then we weighed its wet weight M_1_, put the carrier in a 5 mL measuring cylinder (with an accuracy of 0.1 mL) and measured its total volume V_1_ with drainage method, and take the average value for three times of experimental measurement, as shown in Formula 1.
ρ = M_1_/V_1_(1)

We calculated the water absorption of SBC-TiO_2_ W [27] from formula 2, that is, we dried the prepared composite carrier to constant weight, recorded it as M_0_, then immersed it in deionized water at 20 °C for 24 hours, took it out, used filter paper to absorb the surface water, weighed its wet weight M_1_ and took the average value of each group three times.
W = (M_1_ − M_0_)/M_0_ × 100%(2)

Formula 3 calculation formula for porosity of composite carrier [28], wherein, before the composite carrier was immersed, the weight was recorded as M_1_, then the composite carrier was immersed in deionized water, after 10 hours, the deionized water on its surface was removed with filter paper and quickly weighed as M_2_, the ratio of the mass of the deionized water absorbed and its density (ρ) was recorded as the volume of the internal pores of the composite carrier, which was measured three times average.
Ε = (M_2_ − M_1_)/ρV × 100%(3)

We cut the prepared SBC-TiO_2_ into about 10 mm × 10 mm × 10 mm cubes, randomly selected 20 carrier cubes, put them in deionized water and mechanically stirred them at room temperature, rotating speed of 1000 R min^−1^ to determine the complete number of carrier cubes and calculated the change of retention rate with time [29]. Each group of samples was measured three times to take the average value.

### 2.5. Photocatalytic Activity Test

Different pH values and initial concentrations (2–20 mg/L) of MB solution (200 mL) were considered; a certain amount of the composite carrier (0.5–2 g) was added to the reactor as a photocatalyst. The reactor (400 mL) was placed in a self-designed UV light (light intensity was 24 W/m^2^) test box for 1 h of absorption under dark conditions to achieve adsorption equilibrium. The light source was then turned on and 4 mL was sampled every hour, with a total photocatalytic time of 6 h. After centrifugation, the samples were filtered using a needle filter (0.22 µm). The absorbance was measured using an ultraviolet-visible spectrophotometer (Shimadzu UV-2501PC, kyoto, Japan) and the removal rate was calculated.

## 3. Experimental Results and Analysis

### 3.1. Influence Analysis of Composite Carrier Performance

#### 3.1.1. Effect of Bagasse Cellulose Mass Fraction

Figure 2a shows the effect of the mass fraction of bagasse cellulose on the retention of the composite carrier. The higher the bagasse cellulose content, the higher the water impact force the composite carrier could withstand. This is because an increase in the content can help improve the compactness of the carrier. Therefore, the retention rate increased with the increase in the bagasse cellulose content.

Figure 2b shows that with the increase in the bagasse cellulose content, the porosity of the composite carrier increases first and then decreases. The maximum porosity was 77.68% for a bagasse cellulose content of 4%. The increased porosity can be mainly attributed to the role of bagasse cellulose solution as a backbone of the foaming material, hindering the diffusion of the bubbles from viscose. It can effectively prevent the cavity generated after foaming from collapsing due to the surrounding non-solidified solution, which was conducive to the formation probability of the cavity [30]. However, when the content of bagasse increased to 5%, the bubbles could not resist the bagasse cellulose, resulting in the tight internal pore structure, so the porosity could not be increased continuously.

In Figure 2c, with the increase in the bagasse cellulose content, the water absorption and wet density of the composite carrier increased gradually. This is because bagasse is a type of natural cellulose; a large number of hydroxyl groups enhance its hydrophilicity. However, when the mass fraction increased to a certain extent, the internal structure of the composite carrier becomes compact and the pore size gradually decreases. Therefore, more water cannot be accommodated.

In summary, the mass fraction of bagasse cellulose should be 4% to ensure that the SBC-TiO_2_ composite carrier has a high retention rate, water absorption, a wet density close to that of water and better pore structure.

#### 3.1.2. Influence of PVA Solution Mass Fraction

Figure 3a shows the effect of PVA mass fraction on the retention of the composite carrier. With the increase in the mass fraction of PVA, the retention rate of the carrier increases significantly. When the mass fractions of the PVA were 4% and 5%, the retention rate was 100%. The strength of the carrier was relatively stable during the experiment.

Figure 3b shows the effect of PVA mass fraction on the porosity of the composite carrier. The porosity increases with the increase in the mass fraction of PVA. The probable reason was that PVA as a reinforcing agent cannot fully support the pore structure of the entire composite carrier at low contents. At high contents, PVA can serve as a reinforcing agent, ensuring that the composite carrier has a stable pore structure.

In Figure 3c, the water absorption increases with the increase in the mass fraction of PVA. When this changes, there was a significant impact on the water absorption of the composite carrier. This could be attributed to the high hydrophilicity of PVA which was caused by the large number of hydroxyl groups it contains. Moreover, because of the cross-linking agent, the hydrogen bond in the molecule was reduced and the hydrophilicity was thus increased [31]. In summary, a PVA mass fraction of 5% was considered the optimum.

#### 3.1.3. Effect of Pore Forming Agent Dosage

Figure 4a shows the effect of pore-forming dose on the retention of the composite carrier. Under the optimum mass fractions of bagasse cellulose and PVA, the amount of pore-forming agent has little effect on the retention rate; the retention rate was 100% under different dosages of the pore-forming agent.

Figure 4b shows the effect of the pore-forming dose on the porosity of the composite carrier. With the increase in the dosage of the pore-forming agent, the porosity of the composite carrier first increases and then decreases. The maximum porosity was 81.45% when the dosage of the pore-forming agent was 20 g. The porosity of the composite carrier increases because the foaming effect increases with the increase in the amount of pore-forming agent. The gradual decrease in the porosity was due to the collapse of the composite carrier during the forming and drying periods because of the excessive pore size, making it difficult to support the composite carrier.

With the increase in the amount of pore-forming agent, the water absorption of the composite carrier increases gradually, but decreases when the amount of pore-forming agent reaches 20 g. The wet density of the composite carrier decreases with the increase in the pore-forming agent dosage. This was because with the increase in the amount of pore-forming agent, the porosity and pore size of the composite carrier increase accordingly, making the cellulose and water molecules in the carrier to form hydrogen bonds, thereby enhancing the water absorption of the composite carrier. However, the macroporous structure formed due to the excess pore-forming agent makes it difficult to support the composite carrier during the forming and drying periods. The collapse makes it difficult to keep the large aperture structure stable.

Through the analysis of the above factors, we conclude that the optimum amount of pore-forming agent was 20 g.

### 3.2. Characterization

Figure 5 shows SEM images of the TiO_2_ and biofilms on the surface and in the core of the SBC-TiO_2_. The images confirm the composite carrier was rough, with obvious pore structures, which was beneficial to the adhesion and growth of micro-organisms. Figure 6 shows the successful aggregation of the micro-organisms inside and outside the carrier.

Figure 7 shows the EDS scanning pattern of the composite carrier surface. It can be seen that there were more Ti and O elements on the surface of the composite carrier after being magnified 100 times and their positions almost overlap. This indicates the successful incorporation of nano-TiO_2_ onto the surface of the composite carrier. In the previous characterization of XPS [32], this view has also been confirmed.

### 3.3. Analysis of Factors Influencing Photocatalytic Degradation Reaction

The factors influencing the photocatalytic reaction include substrate concentration, initial pH value and carrier dosage. To study the reaction law, the concentration after dark adsorption was taken as the initial point for a kinetic analysis. Based on experimental data (Table 1, Table 2, Table 3, Table 4, Table 5 and Table 6), the experiment conforms to the pseudo-first-order reaction kinetic equation.

Figure 8 shows the removal rate and reaction kinetics for different initial concentrations of MB (pH 6, the dosage of composite carrier 2 g). As shown in Figure 8a, the adsorption capacity increases gradually with the increase in the initial concentration and then decreases gradually after reaching a maximum at 10 mg L^−1^ under dark adsorption during the first hour. After 6 h of photocatalytic reaction, the final removal rate decreases with the increase in initial concentration. As shown in Figure 8b, the photocatalytic degradation rate decreases with the increase in the initial concentration. The reaction rate was highest at 2 mg L^−1^, which was 6.65 times higher than that at 20 mg L^−1^.

Figure 9 shows the removal rate and reaction kinetics of MB under different dosages of the composite carrier (MB 10 mg/L, pH 6). Figure 9a shows that the removal rate was highest when the dosage was 2 g in the first hour of dark adsorption, because the number of structural sites of MB that can be adsorbed correspondingly increases with the increase in the dosage. After 6 h of photocatalysis, the final degradation efficiency increases with the increase in the dosage. When the dosage of the composite carrier was 2 g, the highest removal rate was 84.24%. As shown in Figure 9b, when the dosage was 2 g, the maximum reaction constant k was 3.89 × 10^−3^ min^−1^. After the dosage was increased to a certain extent, the reaction rate decreases, which can be explained by the availability of titanium dioxide surface active sites and light penetration in the suspension [33].

Figure 10 shows the removal efficiency and reaction kinetics of MB at different pH (MB 10 mg/L, the dosage of composite carrier 2 g). In Figure 10a, the removal rate of MB was highest at a pH of 6 after 6 h. A strong acid or alkali cannot accelerate the removal of MB. This was because the pH value of the solution affects the absorption of hydroxyl on titanium dioxide [34]. As shown in Figure 10b, the reaction rates at pH = 2, 4, 6 and 9 were 1.81 × 10^−3^, 2.35 × 10^−3^, 3.89 × 10^−3^ and 1.78 × 10^−3^ min^−1^, respectively. This also shows that the photocatalytic oxidation of MB was not conducive to over-acidic and over-alkaline conditions.

## 4. Conclusions

In this study, a SBC-TiO_2_ was prepared using a low-temperature foaming method. The composite carrier prepared under optimal conditions exhibited good water absorption, porosity and water retention, all of which are ideal for microbial loading. Characterization results showed that TiO_2_ was successfully incorporated into the carrier. When the mass concentration of MB was 10 mg L^−1^ at 200 mL, 2 g of the composite carrier was added and the initial pH value of the reaction was maintained at 6, the catalytic effect was best under these conditions and the degradation rate reached 78.91% after 6 h. At the same time, in the previous work, we analyzed the degradation pathway of MB. It was found that MB was decomposed into easily mineralized substances [32]. The carrier exhibited a good photocatalytic performance. This study is expected to help optimize ICPB systems for more economical and effective uses.

We did not perform any experiments on the recovery or recycling of the composite carrier. The next step will be in this direction.

## Figures and Tables

**Figure 1 materials-13-01645-f001:**
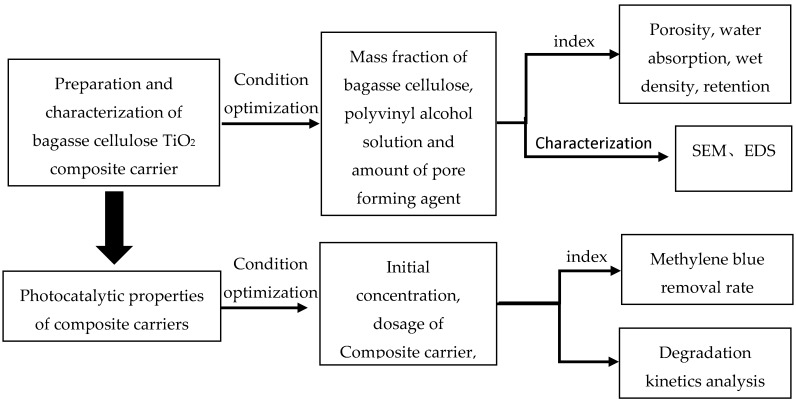
Flow chart of experiment.

**Figure 2 materials-13-01645-f002:**
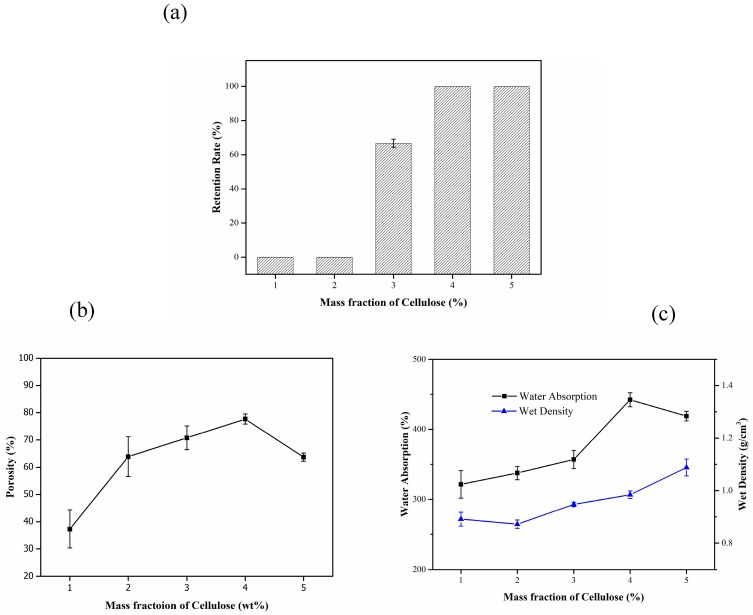
Effect of the mass fraction of cellulose on the properties of the composite carrier. (**a**) Effect of the mass fraction of cellulose on retention rate; (**b**) Effect of the mass fraction of cellulose on porosity; (**c**) Effect of the mass fraction of cellulose on water absorption and wet density.

**Figure 3 materials-13-01645-f003:**
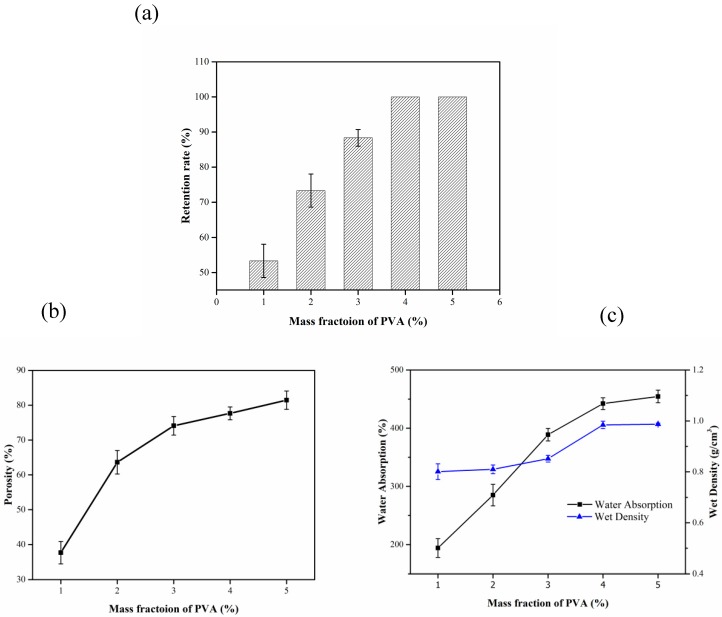
Influence of PVA mass fraction on the performance of composite carrier. (**a**) Effect of the mass fraction of PVA on retention rate; (**b**) Effect of the mass fraction of PVA on porosity; (**c**) Effect of the mass fraction of PVA on water absorption and wet density.

**Figure 4 materials-13-01645-f004:**
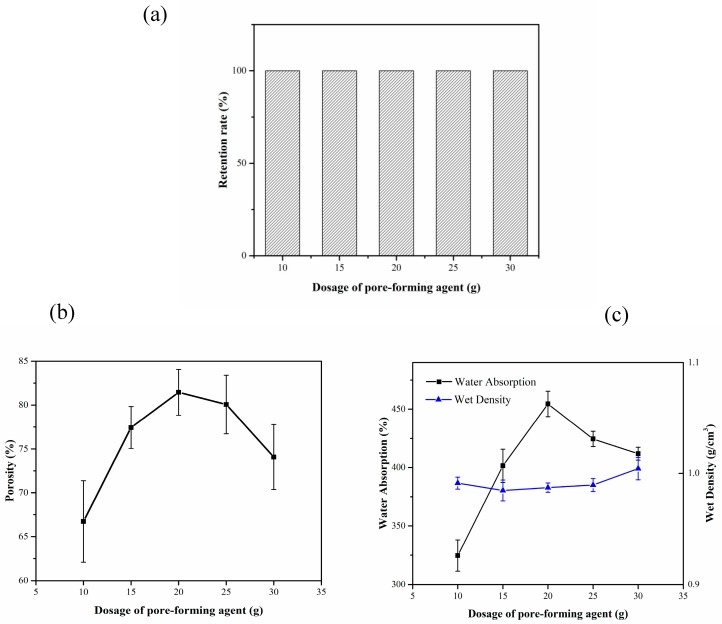
Effect of pore-forming agent dosage on the properties of composite carrier. (**a**) Effect of the dosage of pore-forming agent on retention rate; (**b**) Effect of the dosage of pore-forming agent on porosity; (**c**) Effect of the dosage of pore-forming agent on water absorption and wet density.

**Figure 5 materials-13-01645-f005:**
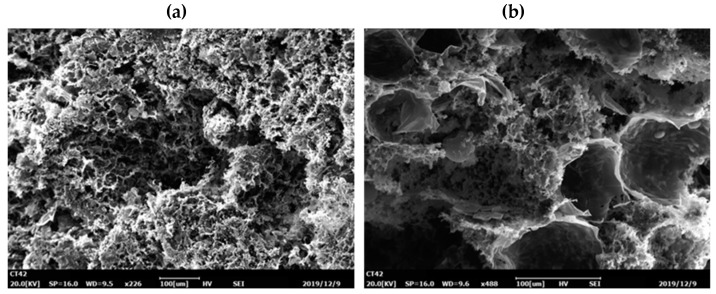
Morphologic characteristics of the surface and interior of the bagasse cellulose-TiO_2_ carrier, obtained using SEM. (**a**) Morphologic characteristics of the surface of the bagasse cellulose-TiO_2_ carrier; (**b**) Morphologic characteristics of the interior of the bagasse cellulose-TiO_2_ carrier.

**Figure 6 materials-13-01645-f006:**
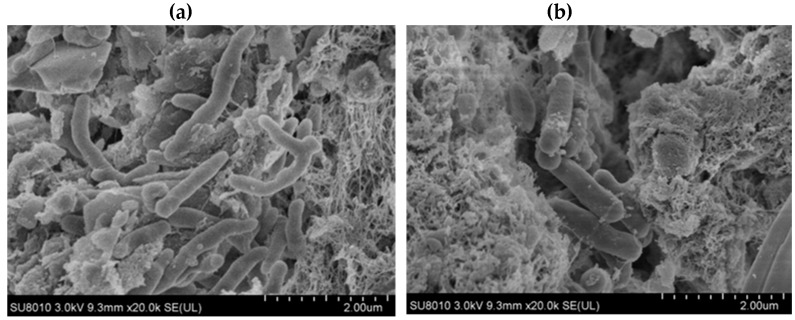
SEM images of an outer surface of the carrier and b interior of the sponge carrier. (**a**) the micro-organisms outside the carrier; (**b**) the micro-organisms inside the carrier.

**Figure 7 materials-13-01645-f007:**
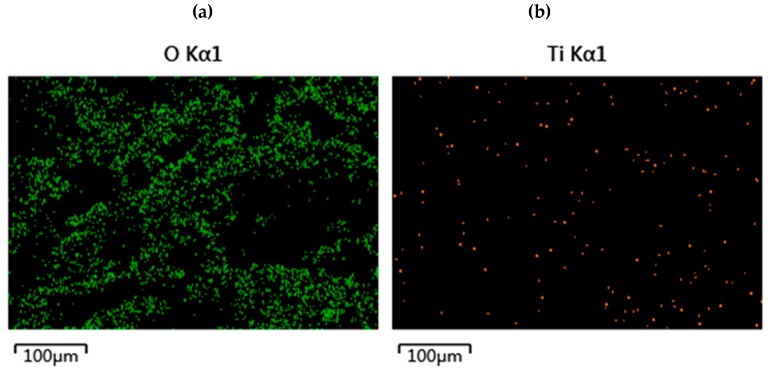
EDS of SBC-TiO_2_ carrier. (**a**) Distribution of O elements on the surface of composite carrier; (**b**) Distribution of Ti elements on the surface of composite carrier.

**Figure 8 materials-13-01645-f008:**
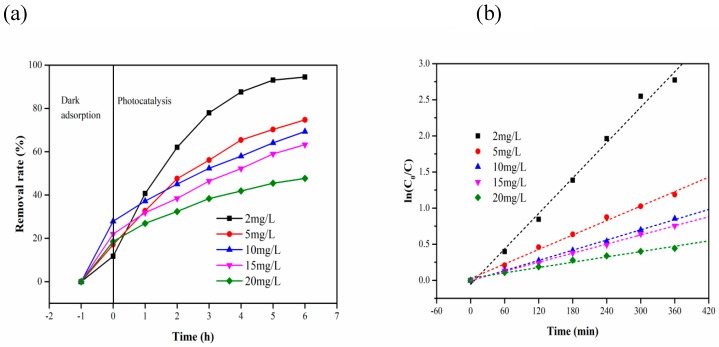
(**a**) Photocatalytic degradation under different initial concentrations of MB; (**b**) kinetics of photocatalysis of MB with different initial concentrations on composite carrier.

**Figure 9 materials-13-01645-f009:**
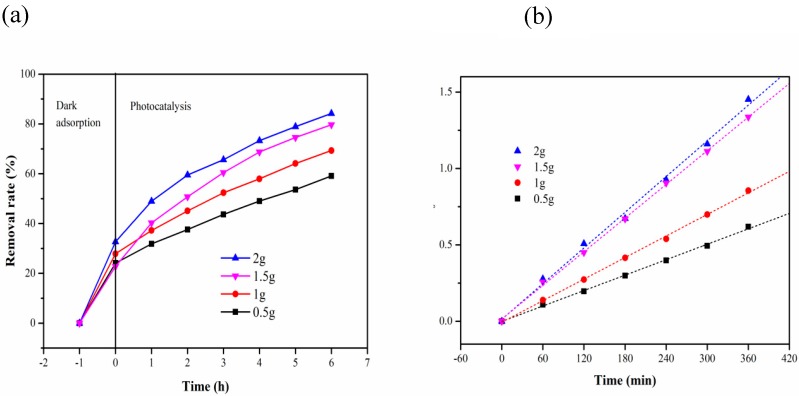
(**a**) Removal rate of MB under different dosages; (**b**) kinetics of photocatalytic degradation of MB with different dosages of the composite carrier.

**Figure 10 materials-13-01645-f010:**
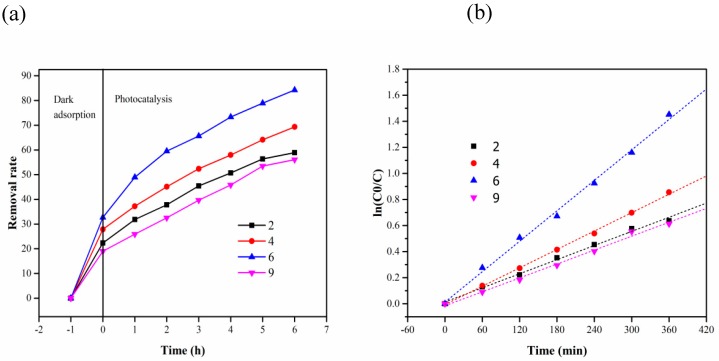
(**a**) Removal rate of MB at different pH; (**b**) kinetics of photocatalytic degradation of MB at different pH.

**Table 1 materials-13-01645-t001:** Photocatalytic degradation rate of MB at different initial concentration.

Time (h) Initial Concentration (mg/L)	2	5	10	15	20
−1	0	0	0	0	0
0	11.72414	17.10037	27.87879	22.02532	18.54083
2	40.68966	32.71375	37.21212	31.81435	26.90763
3	62.06897	47.58364	45.09091	38.39662	32.39625
4	77.93103	56.13383	52.36364	46.49789	38.35341
5	87.58621	65.42751	57.93939	52.23629	41.90094
6	93.10345	70.26022	64.12121	58.98734	45.44846
7	94.48276	74.72119	69.33333	63.29114	47.6573

**Table 2 materials-13-01645-t002:** Reaction kinetics parameters of photocatalytic of MB with different initial concentrations using the SBC-TiO_2_ carrier.

Initial Concentrations	Kinetic Equation	k(min^−1^)	R^2^
2 mg/L	In(C_0_/C) = 0.00818 t − 0.0559	8.18 × 10^−3^	0.9962
5 mg/L	In(C_0_/C) = 0.00334 t + 0.02598	3.34 × 10^−3^	0.9974
10 mg/L	In(C_0_/C) = 0.00235 t − 0.00641	2.35 × 10^−3^	0.9994
15 mg/L	In(C_0_/C) = 0.0021 t − 0.00228	2.1 × 10^−3^	0.9993
20 mg/L	In(C_0_/C) = 0.00123 t + 0.02958	1.23 × 10^−3^	0.9908

**Table 3 materials-13-01645-t003:** Removal rate of MB under different dosage.

Time (h) Carrie Dosage (g)	0.5	1	1.5	2
−1	0	0	0	0
0	24.09	27.88	22.74939	32.68
2	31.87	37.21	40.26764	48.96
3	37.59	45.09	50.72993	59.51
4	43.67	52.36	60.46229	65.63
5	49.03	57.94	68.73479	73.31
6	53.65	64.12	74.57421	78.91
7	59.12	69.33	79.6837	84.24

**Table 4 materials-13-01645-t004:** The reaction kinetics parameters of photocatalytic of MB with different dosage.

Carrier Addition	Kinetic Equation	k (min^−1^)	R^2^
0.5 g	In(C_0_/C) = 0.00168 t − 0.00131	1.68 × 10^−3^	0.9982
1 g	In(C_0_/C) = 0.00235 t − 0.00641	2.35 × 10^−3^	0.9986
1.5 g	In(C_0_/C) = 0.00367 t + 0.01439	3.67 × 10^−3^	0.9992
2 g	In(C_0_/C) = 0.00389 t + 0.014274	3.89 × 10^−3^	0.9954

**Table 5 materials-13-01645-t005:** Removal rate of MB at different pH.

Time (h) pH	2	4	6	9
−1	0	0	0	0
0	22.33129	27.87879	32.68229	18.97507
2	31.82822	37.21212	48.95833	25.90028
3	37.80429	45.09091	59.50521	32.54848
4	45.44785	52.36364	65.625	39.65928
5	50.69325	57.93939	73.30729	45.84488
6	56.35583	64.12121	78.90625	53.4626
7	58.89571	69.33333	84.24479	56.09418

**Table 6 materials-13-01645-t006:** The reaction kinetics parameters of photocatalytic of MB under different pH.

pH	Kinetic Equation	k (min^−1^)	R^2^
2	In(C_0_/C) = 0.00181 t + 0.01402	1.81 × 10^−3^	0.9935
4	In(C_0_/C) = 0.00227 t − 0.00641	2.27 × 10^−3^	0.9916
6	In(C_0_/C) = 0.00398 t + 0.01274	3.98 × 10^−3^	0.9925
9	In(C_0_/C) = 0.00178 t − 0.01478	1.78 × 10^−3^	0.9922

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
