# Peer review of "Preparation and Photocatalytic Properties of a Bagasse Cellulose-Supported Nano-TiO2 Photocatalytic-Coupled Microbial Carrier"

_materials, 2020, doi:10.3390/ma13071645_

Round 1

Reviewer 1 Report

The submitted manuscript presents the properties of a cellulose-supported system representing an intimate coupling of photocatalysis and biodegradation methodology for organic pollutant removal. The developed system consists of photocatalytic material (TiO2), porous carriers (cellulose) and biofilm. Its efficacy was assessed on the basis of methylene blue degradation. Both TiO2 and cellulose carriers were used for this purpose but the employing of bagasse cellulose – fibrous waste of the sugarcane industry – is interesting and worth attention. Unfortunately, I have found it hard to appreciate the idea because of the disorganized fashion of experiment description and data presentation. The description of the conduct of the study is very confusing. I recommend to add a flowchart or scheme describing the separate steps of a process in sequential order and to supplement the missing data (e.g., volumes, concentrations, components ratios, etc. – details are given below).

To assess the contribution of synergic effect of coupling TiO2 and biofilm system for MB degradation, the ability of individual components, TiO2 immobilized in cellulose carrier and biofilm modified cellulose carrier should be tested. 

Next, in the list of references I have not found any review article, especially Intimate coupling of photocatalysis and biodegradation for wastewater treatment: Mechanisms, recent advances and environmental applications; Water Research, https://doi.org/10.1016/j.watres.2020.115673 published in 2020, which covers similar topics. Could the Authors refer to this paper in the present work?

Finally, there are more detailed remarks, which are listed underneath.

  • The process for the production of the cellulose-based carrier needs rewriting. According to the manuscript, the essential components needed are as follows: ZnCl2, bagasse cellulose, polyvinyl alcohol (PVA), titanium dioxide suspension, glutaraldehyde, Na2SO4. The dosage of three of them was optimized (mass fraction of cellulose, 1 – 5%; PVA, 1-5%; Na2SO4, 10-30g). However, the information about the amounts of the remaining ones is missing.
  • ‘A solution containing 100 g of ZnCl2 and bagasse cellulose with a mass fraction of 72% was stirred’ – what was the volume of this mixture? 72% - what do the Authors refer to?
  • ‘10 mL of titanium dioxide suspension was added’ – what was the content of titanium dioxide in this suspension?
  • ‘5 g of the bagasse cellulose titanium dioxide composite carrier’ - In what form was the carrier used? Monolithic or smashed to a powder?
  • ‘The carrier was then transferred to the reactor, a certain concentration of methylene blue solution containing the culture medium was added, and aeration was continued for domestication. The water was changed once a day… After about seven days, the composite carrier was placed in the reactor to carry out a degradation experiment with methylene blue.’ – according to this description the methylene blue (MB) was added twice. The first addition of MB was performed during biofilm formation and the second when the finished product, namely cellulose carrier comprising TiO2 and biofilm, was tested. This should be clarified.
  • ‘The factors influencing the photocatalytic reaction include substrate concentration, initial pH value, and carrier dosage’ -  data shown in Figure 7 was obtained at various concentrations of MB but the information about pH and carrier dosage was not reported. There is a similar situation as far as Figures 8 and 9 are concerned
  • ‘Characterization results showed that TiO2 was successfully incorporated into the carrier. After 6 h of reaction, the methylene blue removal rate reached 78.91%.’ Under what conditions was this value obtained? It is recommended to provide the information on the composition of the carrier, MB concentration and pH.
  • ‘After enlarging the image 100 times we found more Ti and O elements, with largely overlapping positions, on the surface’ – this sentence needs editing
  • ‘After centrifugation, the samples were filtered using a needle filter (0.22 μm)’ – was it a needle filter or a classic syringe one?
  • The Authors do not use subscript or superscript in the chemical formulas and units (e.g., TiO2, ZnCl2, Na2SO4, mg L-1, 3.89 × 10−3 min−1). This must be corrected.

Reviewer 2 Report

Dear Authors,
The present manuscript deals with a quiet interesting subject regarding baggase cellulose-supported nano-TiO2 based materials. The main idea of the manuscript is interesting, in the opinion of the Reviewer, it could be published after addressing some issues. See below some of my concerns regarding the manuscript in the Comments section. Also, I wish all the best for the authors and I hope that the manuscript will be acceptable for publication after revision and improvements.

Best regards,
The Reviewer

0) Some minor typos can be found in manuscript (e.g. double spaces, non-subscript numbers as ZnCl2 or other materials), check them carefully!
1a) Some relevant info you can say about your photocatalytic reactor,e.g. sourse of illumination, wavelenght spectra, construction of the reactor, type of light source etc!
1b) (row 143) "The absorbance was measured using a (...) spectrophotometer"...which one? Some details?
2) In row 154-157 you say that the porosity of the carrier increases first and then decreases, giving explanation only for the increased level of porosity but not for the decreasing, as I've understood. What happens in the second case?
3) I'm completely unsure, if Figure 3a needs to be inserted. I think one sentence is enough, no graph is needed (maybe if some differences are visible, if you "zoom in" at an y scale from 95 to 102% for example)!
4a) At the photocatalytic reactions you have let dark adsorption to make its effect in one your...but as I'm looking on the graph, I'm unsure if it reached its maximum, in order to not effect the evaluation and the evolution of real photocatalytic efficiencies, especially if you are looking closer to the kinetics!
4b) A photolysis investigations of the model contaminant would make the evaluation better, please provide it!

Reviewer 3 Report

This manuscript reports the preparation and photocatalytic properties of a bagasse cellulose-supported Nano-TiO2 photocatalytic-coupled microbial carrier and its application in the methylene blue removal.

The authors characterized the composites by SEM and EDS. However, I think it would be important to complete characterization by using different techniques such as XPS and TG. Besides that, I think that in this case the authors should also supply the surface area and pore size of materials based on BET studies.

In addition, it would also be interesting to see some characterization of the catalytic material after the catalytic reactions. Did a leaching process take place?

Authors should clarify in the abstract which are the best catalytic results obtained.

In section 2.5, authors should indicate the concentrations used for the substrate and for the catalyst or at least their proportions.

The data in figures 7-9 should appear in tables.

In addition, the author must be very clear in the introduction about the novelty of the work.

Authors must improve the manuscript so that it can be accepted.

Round 2

Reviewer 2 Report

Dear Authors, 

Thank you for the answers, I think, the manuscript can be accepted to publication in the present journal. 

Best regards, 
The Reviewer 

Author Response

Dear Reviewer,

       Thank you very much for your valuable comments on our manuscript. Your comments make our manuscripts more rigorous and perfect, so that our manuscripts are more likely to be accepted by journals. Thank you again for your help.

Best regards,

Jianhua Xiong

Reviewer 3 Report

The authors added a comment regarding XPS data. However, concerning other studies requested, namely, the BET studies and the characterization of the catalytic material after the catalytic reactions (to confirm any leaching process). The authors justify the impossibility of carrying out the tests at this stage. I understand. Considering this fact, I have no additional comments at this stage and I want to be neutral in the judgment of this manuscript. I suggest that the editor consider the opinions of other referees.

Author Response

Dear Reviewer,

       Thank you very much for your understanding and the review of our work. We are truly sorry for the impossibility of supplying experiment data at this special moment. We will still try our best to confirm those leaching process by citing corresponding references or other ways, ensuring the rigor of the manuscript.

Sincerely yours,

Jianhua Xiong